# Dihydromyricetin Improves Endothelial Dysfunction in Diabetic Mice via Oxidative Stress Inhibition in a SIRT3-Dependent Manner

**DOI:** 10.3390/ijms21186699

**Published:** 2020-09-13

**Authors:** Yu-Yun Hua, Yue Zhang, Wei-Wei Gong, Yue Ding, Jie-Ru Shen, Hua Li, Yun Chen, Guo-Liang Meng

**Affiliations:** Department of Pharmacology, School of Pharmacy, Key Laboratory of Inflammation and Molecular Drug Target of Jiangsu Province, Nantong University, Nantong 226001, China; 1823310019@stmail.ntu.edu.cn (Y.-Y.H.); 1823310002@stmail.ntu.edu.cn (Y.Z.); 1923310029@stmail.ntu.edu.cn (W.-W.G.); 1823310007@stmail.ntu.edu.cn (Y.D.); 1727011041@stmail.ntu.edu.cn (J.-R.S.); 1727011042@stmail.ntu.edu.cn (H.L.)

**Keywords:** dihydromyricetin, endothelial dysfunction, oxidative stress, sirtuin 3

## Abstract

Dihydromyricetin (DHY), a flavonoid component isolated from Ampelopsis grossedentata, exerts versatile pharmacological activities. However, the possible effects of DHY on diabetic vascular endothelial dysfunction have not yet been fully elucidated. In the present study, male C57BL/6 mice, wild type (WT) 129S1/SvImJ mice and sirtuin 3 (SIRT3) knockout (SIRT3^-/-^) mice were injected with streptozotocin (STZ, 60 mg/kg/day) for 5 consecutive days. Two weeks later, DHY were given at the doses of 250 mg/kg by gavage once daily for 12 weeks. Fasting blood glucose (FBG) and glycosylated hemoglobin (HbA1c) level, endothelium-dependent relaxation of thoracic aorta, reactive oxygen species (ROS) production, SIRT3, and superoxide dismutase 2 (SOD2) protein expressions, as well as mitochondrial Deoxyribonucleic Acid (mtDNA) copy number, in thoracic aorta were detected. Our study found that DHY treatment decreased FBG and HbA1c level, improved endothelium-dependent relaxation of thoracic aorta, inhibited oxidative stress and ROS production, and enhanced SIRT3 and SOD2 protein expression, as well as mtDNA copy number, in thoracic aorta of diabetic mice. However, above protective effects of DHY were unavailable in SIRT3^-/-^ mice. The study suggested DHY improved endothelial dysfunction in diabetic mice via oxidative stress inhibition in a SIRT3-dependent manner.

## 1. Introduction

Diabetes is a chronic disease characterized by hyperglycemia. The increasing prevalence of diabetes becomes a serious global health problem [1,2,3]. There are various chronic complications of diabetes, which are the main cause of morbidity and mortality of diabetes, including: diabetic nephropathy, neuropathy, retinopathy, lower extremity vascular lesions, and gastroenteropathy [4,5,6,7]. Prolonged hyperglycemia may impair endothelial function of macro- and micro-vascular to increase risks for cardiovascular disease in diabetes [8,9]. Therefore, it is of great importance to improve endothelial function in diabetes.

The detailed molecular mechanisms involved in endothelial dysfunction with diabetes are complex. Oxidative stress is considered to be one key factor causing diabetic vascular endothelial dysfunction [10,11,12,13]. Increased reactive oxygen species (ROS) accumulation in the mitochondria of coronary artery endothelial cells was found in diabetic rats [12]. Previous studies verified that antioxidants and superoxide dismutase restored endothelium-dependent vasodilation of streptozocin (STZ)-induced diabetic rats [14]. ROS scavenger N-acetyl-L-cysteine (NAC) pretreatment alleviated high glucose (HG)-induced endothelial dysfunction in human microvascular endothelial cell-1 (HMEC-1) cells [15]. Above evidence suggested the utility of antioxidants for diabetic vascular endothelial dysfunction.

Dihydromyricetin (DHY) is the richest and most bioactive flavonoid component from *Ampelopsis grossedentata* (Figure 1). DHY exhibits versatile pharmacological activities such as anti-inflammatory, anticancer, anti-oxidative, neuroprotective, and cell death-mediating activities [16,17,18]. DHY has been demonstrated to significantly reduce ROS production and malondialdehyde (MDA) but preserve the activities of superoxide dismutase (SOD), catalase (CAT), and Glutathione peroxidase (GSH-Px) in high fat fed middle-aged LDL receptor knockout (LDL^-/-^) mice [19]. Moreover, DHY administration attenuated oxidative stress in Ang II-induced cardiomyocyte hypertrophy [20]. Additionally, DHY protected human umbilical vein endothelial cells (HUVECs) from hydrogen peroxide (H_2_O_2_)-induced oxidative stress damage [21]. However, the possible effects of DHY on diabetic vascular endothelial dysfunction have not yet been fully elucidated.

Sirtuin 3 (SIRT3) is the main deacetylase in the mitochondria and serves as a critical mediator in metabolic regulation and antioxidant function [22,23,24,25]. SOD2, as one target protein of SIRT3, is regulated by SIRT3 via acetylation of lysine residues [26,27]. Transgenic mice with global SIRT3 overexpression improved endothelial dysfunction and attenuated vascular oxidative stress, while SIRT3 depletion promoted endothelial dysfunction and exacerbate vascular hypertrophy in hypertension [28]. It is worth noting that our previous study found DHY attenuated transverse aortic constriction induced myocardial hypertrophy through SIRT3 enhancement [18].

In the current study, we aimed to investigate the effects and the possible mechanism of DHY on endothelial dysfunction in diabetic mice. It is beneficial to provide a novel strategy for the prevention and treatment of vascular endothelial dysfunction in diabetes.

## 2. Results

### 2.1. DHY Decreased FBG and HbA1c Level in Diabetic Mice

During the whole experiment, FBG level remained more than 16.7 mmol/L in diabetic mice, confirming the successful establishment of diabetic mice model. In addition, diabetic mice displayed a marked increase in HbA1c level compared to mice in the control group. It is noting that there was significant decline in FBG level in diabetic mice with DHY administration. Similarly, there was a significant reduction in HbA1c level in diabetic mice after DHY treatment (Figure 2).

### 2.2. DHY Improved Endothelium-Dependent Relaxation of Thoracic Aorta in Diabetic Mice

The endothelial function of aortae was examined after DHY treatment by evaluating acetylcholine (Ach)-induced endothelium-dependent relaxation [29]. The results revealed that ACh-induced endothelium-dependent relaxation in mouse aortae was impaired in diabetic mice, which was significantly improved by DHY treatment. However, there was no significant difference on relaxation induced by sodium nitroprusside (SNP) among the four groups (Figure 3). These data demonstrated that DHY ameliorated impaired endothelium-dependent relaxation of thoracic aorta in diabetic mice.

### 2.3. DHY Suppressed Oxidative Stress in Thoracic Aorta of Diabetic Mice

To investigate the contribution of oxidative stress in the progress of vascular dysfunction in diabetic mice and the effects of DHY on oxidative stress in diabetic mice, the oxidative status was assessed by determining MDA level and the anti-oxidant biomarkers including total antioxidant capacity (T-AOC) activity, SOD activity, and glutathione/glutathione disulfide (GSH/GSSG) level. Dihydroethidium (DHE) staining, 2′, 7′-dichlorofluorescein diacetate (DCFH-DA) staining and MitoSOX staining were used to measure superoxide level, ROS production and mitochondrial superoxide level respectively. Diabetic mice showed increased MDA level superoxide level, ROS production and mitochondrial superoxide level, but decreased T-AOC activity, SOD activity (mainly SOD2 in mitochondria, but not SOD1 in the cytoplasm), and GSH/GSSG level compared with the control group, and DHY treatment significantly suppressed global cellular and mitochondrial oxidative stress in thoracic aorta of mice with diabetes (Figure 4 and Figure 5).

### 2.4. DHY Enhanced SIRT3 and SOD2 Protein Expression in Thoracic Aorta of Diabetic Mice

Previous studies have revealed close relation among SIRT3, oxidative stress and endothelial function [28,30]. Furthermore, SOD2, an important substrate of SIRT3, has powerful activity to suppress ROS production. As shown in Figure 6, SIRT3 and SOD2 expression were obviously reduced in thoracic aorta of diabetic mice, which were reversed with DHY treatment.

### 2.5. DHY Failed To Improve Endothelium-Dependent Relaxation of Thoracic Aorta in SIRT3^-/-^ Mice with Diabetes

To investigate the role of SIRT3 in the protective effects of DHY on endothelium-dependent relaxation of thoracic aorta, SIRT3^-/-^ mice were further used in our study. The study showed that DHY improved endothelium-dependent relaxation of thoracic aorta in wild type (WT) diabetic mice, which was abolished in SIRT3^-/-^ mice with diabetes (Figure 7A). There was no significant difference on relaxation induced by SNP among the four groups (Figure 7B).

### 2.6. DHY Failed to Suppress Oxidative Stress of Thoracic Aorta in SIRT3^-/-^ Mice with Diabetes

As shown in Figure 8 and Figure 9A, DHY attenuated MDA level, superoxide level, and ROS production, as well as mitochondrial superoxide level, but enhanced T-AOC activity, GSH/GSSG level, and SOD2 activity in the thoracic aorta of WT mice with diabetes. However, DHY treatment had no significant effect on the above indexes of SIRT3^-/-^ diabetic mice.

SIRT3 is localized in mammalian mitochondria and deficiency of SIRT3 is detrimental to mitochondrial function. Mitochondrial oxidative stress leads to mtDNA damage and lower mtDNA copy number indicates serious mitochondrial dysfunction, contributing to endothelial dysfunction [31,32]. The results showed decreased mtDNA copy number in thoracic aorta of WT diabetic mice, which was markedly increased after DHY treatment. However, DHY treatment had no significant effect on the mtDNA copy number of SIRT3^-/-^ diabetic mice (Figure 9B). These findings indicated that DHY ameliorated diabetic vascular endothelial dysfunction in a SIRT3-dependent manner.

## 3. Discussion

The diabetes with an increasing morbidity has become a worldwide public problem threatening to human health [33,34]. According to the 2018 International Diabetes Federation, about 425 million adults worldwide suffered from diabetes, and the number is expected to exceed 592 million in the next 25 years [35]. Diabetes is closely related to cardiovascular diseases, leading to organ failure and premature death [36,37,38]. STZ-induced diabetes is a widely used experimental model of type 1 diabetes. Islet cells are selectively destroyed by STZ, characterized by insulin level lowering and blood glucose level elevating [39,40,41]. In our study, the FBG and HbA1c level were increased obviously in STZ-induced diabetic mice. Consistent with previous studies, DHY exerted the hypoglycemic effect in the present study, as evidenced by the decrease in FBG and HbA1c level of STZ-induced diabetic mice. It has been verified that DHY can improve glucose metabolism and insulin sensitivity and ameliorate insulin resistance not only in experimental studies but also in terms of clinical efficacy [42,43,44]. DHY has been also proven to attenuate atherosclerosis via improving endothelial dysfunction or through microRNA-21 [45,46] and to protect against ligation-induced carotid artery neointimal formation [47]. In addition, DHY attenuated TNF-α-induced endothelial dysfunction [48]. Taken together, DHY improved vascular endothelial function. In our experimental conditions, considerable decrease in the endothelial-dependent relaxation to Ach indicated impaired endothelial function in diabetic mice. Importantly, our results did show that DHY improved endothelium function in the thoracic aortas of STZ-induced diabetic mice. The improvement of endothelial function suggested that NO production increased by DHY. However, changes in NO expression and nitrative stress had not been measured in the present study.

The above vascular protective effects of DHY might attribute to its anti-oxidative ability. Previous evidence has suggested that DHY inhibited oxidative stress in the hippocampus to improve the cognitive impairment of type 2 diabetes mellitus (T2DM) mice [49]. The anti-oxidative properties of DHY were also reported to involve in protecting human umbilical vein endothelial cells from hydrogen peroxide induced oxidative stress damage [21]. Our previous studies have revealed that DHY suppressed Ang II induced cardiac fibroblast proliferation by decreasing cellular production of ROS and MDA level and increasing the SOD activity and T-AOC [50]. Furthermore, DHY pretreatment decreased myocardial ROS production and MDA level, while increased T-AOC, SOD activity and SOD2 expression to ameliorate transverse aortic constriction (TAC) induced myocardial hypertrophy [18]. Similarly, our current study found that DHY reduced the level of MDA and GSH/GSSG and alleviated DHE and DCFH-DA fluorescence intensity, but it enhanced T-AOC activity. Accumulating evidence has elucidated mitochondrial oxidative stress is closely associated with endothelial dysfunction [32,51,52]. Our results showed decreased SOD2 activity and expression, weak MitoSOX fluorescence intensity, as well as reduced mtDNA copy number, in the thoracic aorta of STZ-induced diabetic mice, which were restored by DHY. A variety of researches have also confirmed that stalling of vascular oxidative stress contributes to improvement of endothelial function [53,54,55,56,57,58]. All these data suggested that DHY possibly improved the endothelium-dependent relaxation via its anti-oxidative ability.

The cardioprotective effect of DHY prevents ischemia-reperfusion-induced apoptosis in vivo and in vitro via the PI3K/Akt and HIF-1α signaling pathways [59]. Further study demonstrated that DHY treatment increased SIRT3 level in the heart of myocardial ischemia/reperfusion mice [60]. DHY also activated SIRT3 to suppress chondrocytes degeneration [61]. Similarly, DHY increased SIRT3 expression to ameliorate nonalcoholic fatty liver disease [62]. SIRT3 mediated DHY-induced amelioration of hepatic steatosis and oxidative injury by improving mitochondrial functions [62]. In line with these observations, the present study revealed that DHY increased SIRT3 expression in the thoracic aorta of diabetic mice. We used the SIRT3^-/-^ mice to establish a diabetic model to confirm the contribution of SIRT3 to DHY on diabetic vascular endothelial dysfunction. We further demonstrated that the protective effects of DHY on diabetic vascular endothelial dysfunction were abolished in SIRT3^-/-^ mice. Altogether, DHY improved endothelium-dependent relaxation of thoracic aorta in diabetic mice via a SIRT3 dependent manner.

In conclusion, the present data indicated that DHY improved endothelium-dependent relaxation of the diabetic aorta, mainly through alleviating oxidative stress and activating SIRT3. It suggested DHY may serve as a candidate agent to ameliorate vascular endothelial dysfunction in diabetes.

## 4. Materials and Methods

### 4.1. Animal Treatment

Male 8-week C57BL/6 mice, wild type (WT)129S1/SvImJ mice, and SIRT3 knockout (SIRT3^-/-^) mice were adaptively fed with a normal diet and given free access to water for a week. After a 12-h overnight fast, mice were intraperitoneally injected with streptozotocin (STZ, 60 mg/kg/day, Sigma-Aldrich, St. Louis, MO, USA) in 0.1 mol/L citrate buffer for 5 consecutive days. Some other mice were injected with the same amount of citrate buffer and severed as a control group. After fasting for 8 h, fasting blood glucose (FBG) was measured with a OneTouch glucometer. Mice with FBG level above 16.7 mmol/L were considered as diabetes mellitus (DM). Two weeks after STZ injection, DHY at doses of 250 mg/kg in 0.5% carboxymethylcellulose (CMC) were administrated by gavage for 12 weeks. The same amount of CMC was given to the mice in control group. During the whole experiments, the FBG level was detected every two weeks.

All the procedures were in accordance with both the recommendations of the Guidelines for the Care and Use of Laboratory Animals published by the National Institutes of Health and the Instructional Animal Care and Use Committee of Nantong University (NTU number 20181127, on 27 November 2018).

### 4.2. Measurement of the Glycosylated Hemoglobin (HbA1c)

After treatment for 12 weeks, blood samples were collected and heparinized. The level of HbA1c was measured using an enzyme-linked immunosorbent assay (ELISA) Kit according to the manufacturer’s instructions (Jiancheng, Nanjing, China).

### 4.3. Assessment of Endothelium-Dependent Relaxations

Thoracic aorta of each mouse was detached and connective tissue and excess adipose were removed carefully. The vessels were cut into approximately 4 mm ring segments and then mounted into an organ bath (DMT, Aarhus, Denmark) full of fresh Krebs solution at 37 °C with 95% O_2_ and 5% CO_2_. The rings were placed in a force transducer and stretched to a testing tension of 9.8 mN throughout the experiment. After an equilibration of 1 h, the aortic segments were pre-contracted with noradrenaline (NE, Jinrao amino acid company, Tianjin, China, 10^−7^ mol/L). When a stable plateau was reached, endothelium-dependent or independent relaxation was measured by testing the concentrations-response relation upon the cumulative addition of acetylcholine (Ach, 10^−9^ to 10^−5^ mol/L; Sigma-Aldrich, St. Louis, MO, USA) or sodium nitroprusside (SNP) (10^−10^ to 10^−6^ mol/L; Sigma-Aldrich, St. Louis, MO, USA). The relaxation at each concentration was measured and expressed as the percentage of force generated in response to NE.

### 4.4. Biochemaical Assessment

The MDA level in the thoracic aorta was detected with a lipid peroxidation assay kit according to the manufacture’s instruction (Beyotime, Shanghai, China). Total antioxidant capacity (T-AOC) of the thoracic aorta was assessed with a T-AOC assay kit using a rapid 2,2′-azino-bis 3-ethylbenzthiazoline-6-sulfonic acid (ABTS) method (Beyotime, Shanghai, China). The glutathione/oxidized glutathione (GSH/GSSG) level was determined with a GSH/GSSG assay kit according to the manufacture’s instruction (Beyotime, Shanghai, China). Activity of the superoxide dismutase (SOD) of the thoracic aorta was detected with SOD Assay Kit using the WST-1 method (Beyotime, Shanghai, China).

### 4.5. Measurement of Superoxide Formation in the Thoracic Aorta

Tissue-Tek optimal cutting temperature (OCT) (Sakura Finetek, Tokyo, Japan) embedded frozen sections of thoracic aorta (5 µm thickness) were incubated with dihydroethidium (DHE, 2 µM, Beyotime, Shanghai, China) for 30 min at 37 °C in dark. The slides were morphometrically photographed with a laser confocal microscope (Leica, Wetzlar, Germany).

### 4.6. Measurement of Intracellular ROS

The level of intracellular ROS was measured by using the Reactive Oxygen Species Assay Kit according to the manufacturer’s instructions (Beyotime, Shanghai, China). Briefly, sections of thoracic aorta (5 µm thickness) were incubated with DCFH-DA, and kept in the dark for 20 min at 37 °C. The slides were morphometrically photographed with a laser confocal microscope (Leica, Wetzlar, Germany).

### 4.7. Measurement of Mitochondrial ROS

The level of intracellular ROS was measured by using the Mitochondrial Superoxide Indicator according to the manufacturer’s instructions (Yeasen, Shanghai, China). Briefly, sections of thoracic aorta (5 µm thickness) were incubated with MitoSOX and kept in the dark for 10 min at 37 °C. The slides were morphometrically photographed with a laser confocal microscope (Leica, Wetzlar, Germany).

### 4.8. Western Blot

Protein samples were separated with sodium dodecyl sulfate polyacrylamide gel electrophoresis (SDS-PAGE) and transferred to polyvinylidene difluoride (PVDF) membrane (Millipore, Billerica, MA, USA). After blocking, the primary antibodies were incubated overnight at 37 °C followed by HRP-conjugated secondary antibodies incubation for 2 h. The primary antibodies used were listed as follows: anti-SIRT3 (1:1000, Santa Cruz Biotechnology Inc., San Diego, CA, USA), anti-SOD2 (1:1000, Abcam, Cambridge, UK), and anti-β-tubulin (1:3000, Bioworld Technology, St. Louis, MO, USA). Protein bands were visualized with enhanced chemiluminescence (ECL) (Thermo Fisher Scientific Inc., Rockford, IL, USA). The target protein expression level was normalized to the level of β-tubulin.

### 4.9. Measurement of mtRNA Copy Number

The ratio mtDNA versus nuclear DeoxyriboNucleic Acid (nDNA) was measured using a quantitative real-time PCR assay. The primers were designed to target nDNA (β-actin, 5′-ATGGTGGGAATGGGTCAGAA-3′ and 5′-CTTTTCACGGTTGGCCTTAG-3′) or mtDNA (NADH dehydrogenase 1, 5′ -AAACGCCCTAACAACCAT-3′ and 5′-GGATAGGATGCTCGGATT-3′). The mtDNA content was normalized to the expression of β-actin gene to calculate the relative mtDNA copy number. Each measurement was repeated in triplicate, and a non-template control was included in each experiment.

### 4.10. Statistical Analysis

All data were presented as standard error of the mean (S.E.M.). Statistical analysis was performed by one-way ANOVA test and followed by Bonferroni post hoc test on comparisons among multiple groups. *p* < 0.05 was considered statistically significant.

## Figures and Tables

**Figure 1 ijms-21-06699-f001:**
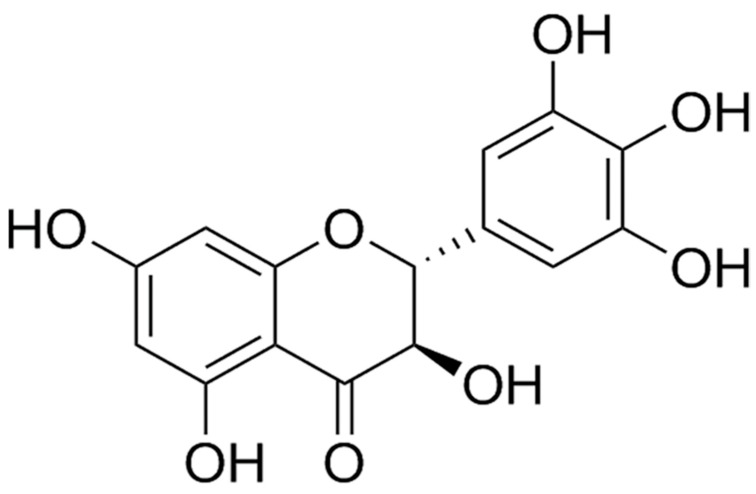
Structure of dihydromyricetin (DHY).

**Figure 2 ijms-21-06699-f002:**
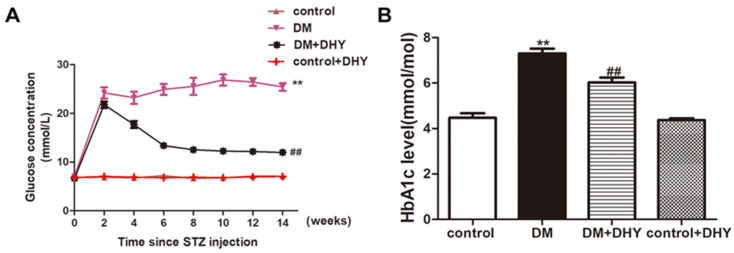
Effects of DHY on fasting blood glucose (FBG) and glycosylated hemoglobin (HbA1c) level in mice. Male C57BL/6 mice were injected with 60 mg/kg/day streptozotocin (STZ) for 5 consecutive days. Mice in control group were injected the same amount of citrate buffer. After 2 weeks, DHY at doses of 250 mg/kg or carboxymethylcellulose (CMC) (0.5%) were given once daily by gavage for 12 weeks. (**A**) FBG level was measured periodically from mice in each group every 2 weeks. (**B**) HbA1c level was detected in mice from each group. ** *p* < 0.01 versus control, ^##^
*p* < 0.01 versus diabetes mellitus (DM) (*n* = 6).

**Figure 3 ijms-21-06699-f003:**
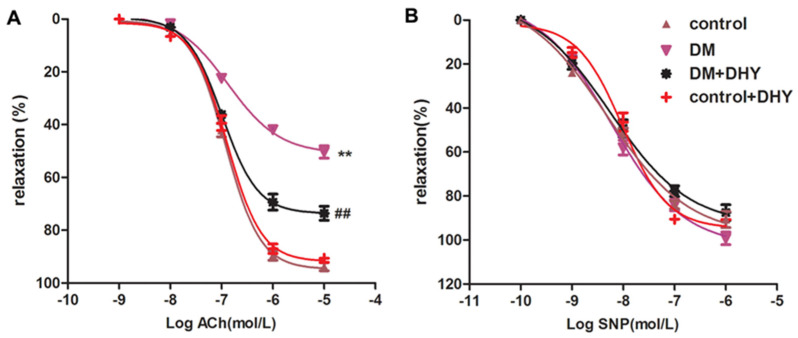
Effects of DHY on endothelium-dependent relaxation of thoracic aorta in mice. (**A**) Ach-induced endothelium-dependent relaxation of thoracic aortae was evaluated from mice in each group. (**B**) Sodium nitroprusside (SNP)–induced relaxation of thoracic aortae was evaluated from mice in each group. ** *p* < 0.01 versus control, ^##^
*p* < 0.01 versus DM (*n* = 6).

**Figure 4 ijms-21-06699-f004:**
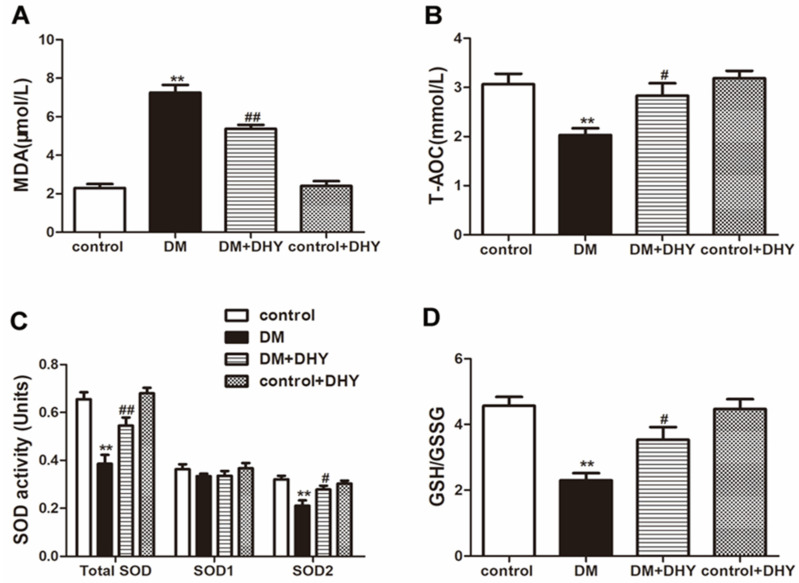
Effects of DHY on oxidative stress in thoracic aorta of diabetic mice. (**A**–**D**) Malondialdehyde (MDA) level, total antioxidant capacity (T-AOC) activity, superoxide dismutase (SOD) activity and glutathione/glutathione disulfide (GSH/GSSG) level in thoracic aorta were measured. ** *p* < 0.01 versus control, ^#^
*p* < 0.05, ^##^
*p* < 0.01 versus DM (*n* = 6).

**Figure 5 ijms-21-06699-f005:**
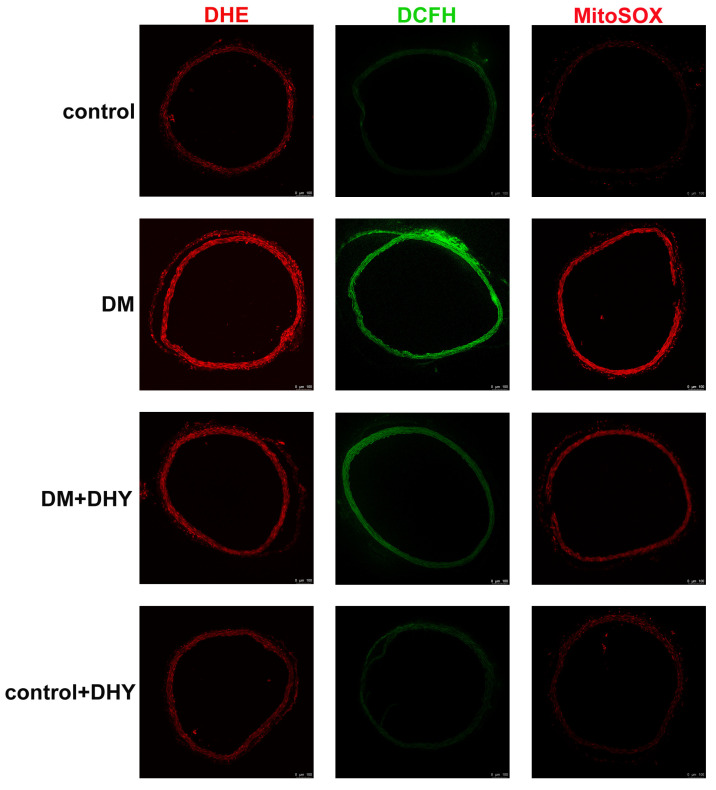
Effects of DHY on reactive oxygen species (ROS) production in thoracic aorta of diabetic mice. Representative images of dihydroethidium (DHE) staining (Red), 2′, 7′-dichlorofluorescein diacetate (DCFH-DA) staining (Green), and MitoSOX staining (Red) of the thoracic aorta are shown. Bar = 100 μm.

**Figure 6 ijms-21-06699-f006:**
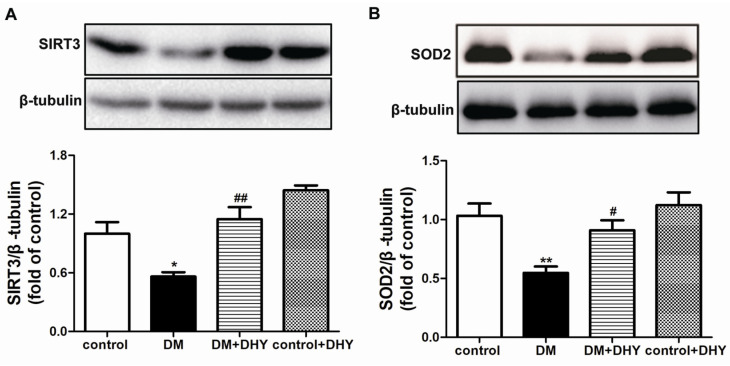
Effects of DHY on sirtuin 3 (SIRT3) and SOD2 expression in thoracic aorta of diabetic mice. (**A**–**B**) SIRT3 expression and SOD2 in thoracic aorta was detected with western blot. β-tubulin was used as a loading control. * *p* < 0.05, ** *p* < 0.01 versus control, ^#^
*p* < 0.05, ^##^
*p* < 0.01 versus DM (*n* = 6).

**Figure 7 ijms-21-06699-f007:**
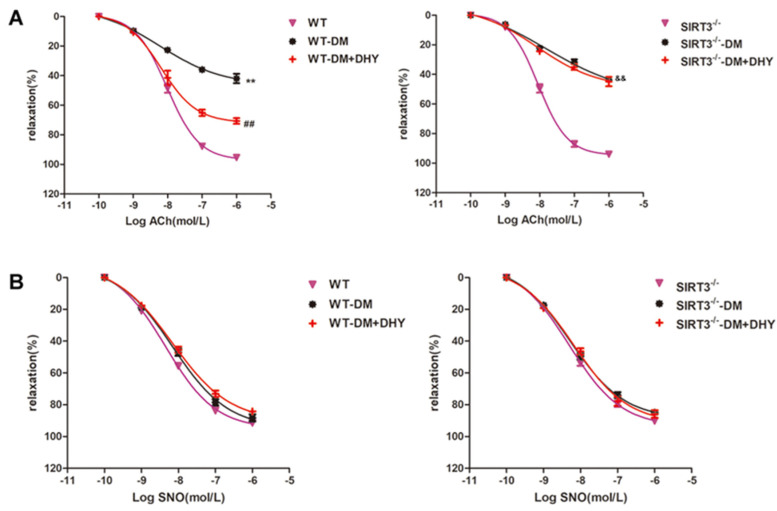
Effects of DHY on endothelium-dependent relaxation of thoracic aorta in SIRT3^-/-^ diabetic mice. Male 129S1/SvImJ mice and male SIRT3 knockout mice (SIRT3^-/-^) were injected with 60 mg/kg/day STZ for 5 consecutive days. Mice in control group were injected the same amount of citrate buffer. After 2 weeks, DHY at doses of 250 mg/kg or carboxymethylcellulose (CMC) (0.5%) were given once daily by gavage for 12 weeks. (**A**) Ach-induced endothelium-dependent relaxation of thoracic aortae was evaluated from mice in each group. (**B**) SNP-induced relaxation of thoracic aortae was evaluated from mice in each group. ** *p* < 0.01 versus wild type (WT), ^##^
*p* < 0.01 s versus WT-DM, ^&&^
*p* < 0.01 versus SIRT3^-/-^ (*n* = 6).

**Figure 8 ijms-21-06699-f008:**
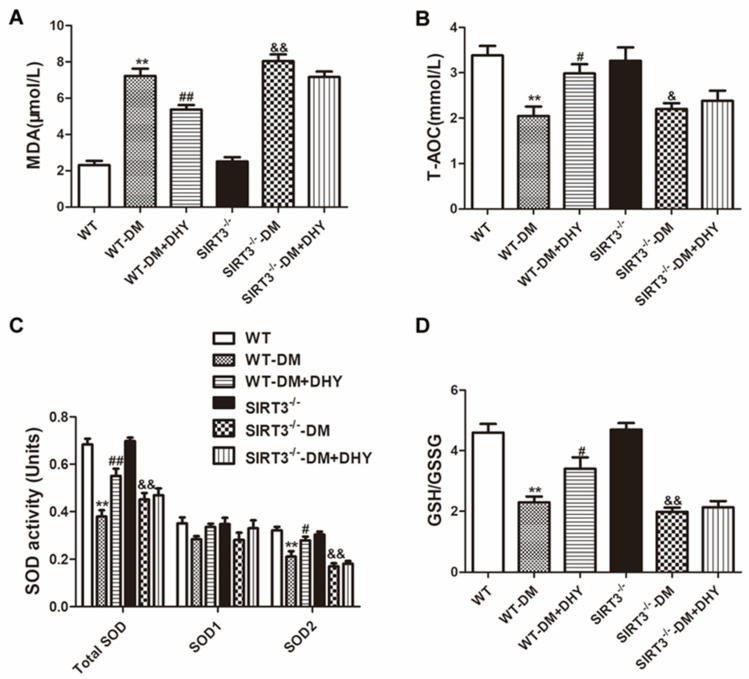
Effects of DHY on oxidative stress in thoracic aorta of SIRT3^-/-^ diabetic mice. (**A**–**D**) MDA level, T-AOC activity, SOD activity, and GSH/GSSG level in thoracic aorta were measured. ** *p* < 0.01 versus WT, ^#^
*p* < 0.05, ^##^
*p* < 0.01 versus WT-DM, ^&^
*p* < 0.05, ^&&^
*p* < 0.01 versus SIRT3^-/-^ (*n* = 6).

**Figure 9 ijms-21-06699-f009:**
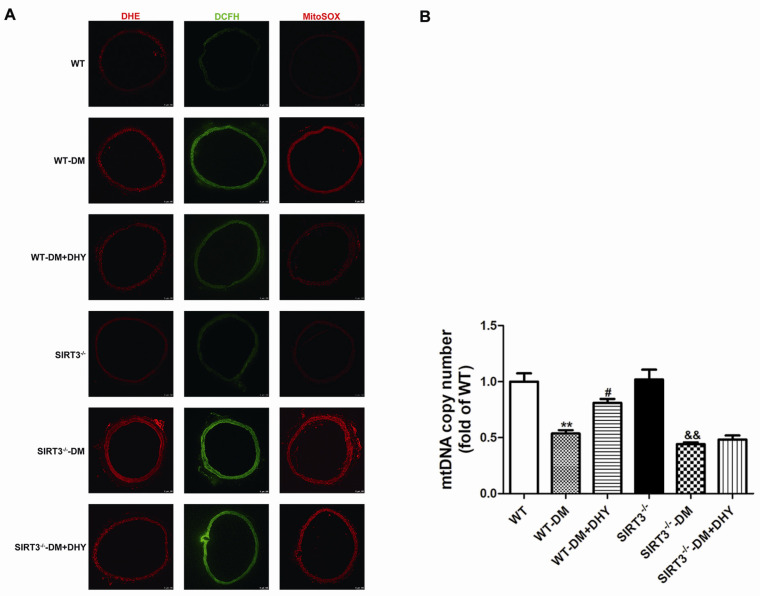
Effects of DHY on oxidative stress in thoracic aorta of SIRT3^-/-^ diabetic mice. (**A**) Representative images DHE staining (Red), 2′, 7′-dichlorofluorescein diacetate (DCFH-DA) staining (Green), and MitoSOX staining (Red) of the thoracic aorta were shown. The nuclei were counter-stained with 2-(4-Amidinophenyl)-6-indolecarbamidine dihydrochloride (DAPI) (Blue). Bar = 100 μm. (**B**) Mitochondrial Deoxyribonucleic Acid (mtDNA) copy number was measured by real-time PCR. ** *p* < 0.01 versus WT, ^#^
*p* < 0.05 versus WT-DM, ^&&^
*p* < 0.01 versus SIRT3^-/-^ (*n* = 6).

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
