# Peer review of "Dihydromyricetin Improves Endothelial Dysfunction in Diabetic Mice via Oxidative Stress Inhibition in a SIRT3-Dependent Manner"

_ijms, 2020, doi:10.3390/ijms21186699_

Round 1

Reviewer 1 Report

The article by Hua et al.focuses on the role of DHY in improving endothelial dysfunction and oxidative stress. The goal and presentation of the authors' view appear confusing and can not be accepted in its present format. 

  1. Is the goal of this manuscript is to study mitochondrial oxidative stress of general oxidative stress.
  2. Authors have chosen to evaluate SIRT3 and SOD2, both are mitochondrial. However, the discussion appears very descriptive and doesn't go into the details of these changes in relation to mitochondrial dysfunction. 
  3. What was the basis to choose parameters as mentioned in Figs. 4 & 8? 
  4. Authors have used a non-specific generalized staining marker of oxidative stress, DHE. This assay does not provide specific details of oxidative stress. It is imperative to include more specific parameters instead. 
  5. What could be the reason for DHE staining even in control sample?
  6. Authors need to perform more mitochondrial related parameters if they are focused on SIRT3.
  7. Did the authors evaluate changes in nitric oxide expression? and particularly, nitrative stress markers? 

Overall, this manuscript requires additional data and overall rewriting to be more focused. 

Author Response

English language and style

( ) Extensive editing of English language and style required  
(x) Moderate English changes required  
( ) English language and style are fine/minor spell check required  
( ) I don't feel qualified to judge about the English language and style 

Thanks very much for your comments. We have double checked the MS to avoid any typographical and grammar mistakes. The MS has also been read and altered by a native English speaker and we hope that it is improved. All the corrections have been highlighted in red in the revised MS.

Yes

Can be improved

Must be improved

Not applicable

Does the introduction provide sufficient background and include all relevant references?

()

( )

(x)

( )

Is the research design appropriate?

( )

(x)

()

( )

Are the methods adequately described?

( )

(x)

( )

( )

Are the results clearly presented?

( )

( )

(x)

()

Are the conclusions supported by the results?

( )

()

(x)

()

Thanks very much for your comments. According to your and other reviewers' suggestion, several additional experiments have been performed to improve the quality of our study. The resultsdescription and conclusion have also been revised.

  1. Is the goal of this manuscript is to study mitochondrial oxidative stress of general oxidative stress.

Response 1: Thanks very much for your comments. We have performed additional experiments to measure ROS production and mitochondrial superoxide level with 2’, 7’-dichlorofluorescein diacetate (DCFH-DA) staining and MitoSOX staining respectively. We found that DHY reduced the level of MDA and GSH/GSSG, alleviated DHE and DCFH fluorescence intensity but enhanced T-AOC activity. It indicated that DHY suppressed general oxidative stress in thoracic aorta of diabetic mice. Furthermore, we also found DHY treatment enhanced SOD activity (mainly SOD2 in mitochondria) and SOD2 expression, but decreased MitoSOX fluorescence, suggesting the inhibition of DHY on mitochondrial oxidative stress. The related information has been added into the the revised MS (page 4, line 5-13, page 5, line 1-4, page 7, line 1-2, page 9, line11-15).

  1. Authors have chosen to evaluate SIRT3 and SOD2, both are mitochondrial. However, the discussion appears very descriptive and doesn't go into the details of these changes in relation to mitochondrial dysfunction. 

Response 2:Thanks very much for your comments. Based on your suggestion, SIRT3 and SOD2 changes in relation to mitochondrial dysfunction have been added into the discussion (page 9, line 12-15)

  1. What was the basis to choose parameters as mentioned in Figs. 4 & 8?

Response 3: Thanks very much for your comments. The basis to choose parameters mentioned in Figs. 4 & 8 including MDA level, T-AOC activity, SOD activity and glutathione/glutathione disulfide (GSH/GSSG) level has been added into the revised MS(page 4, line 5-10).

  1. Authors have used a non-specific generalized staining marker of oxidative stress, DHE. This assay does not provide specific details of oxidative stress. It is imperative to include more specific parameters instead. 

Response 4: Thanks very much for your comments. We have performed additional experiments with DCFH staining and MitoSOX staining for further measurement of oxidative stress. The related information has been added into the the revised MS (page 4, line 5-13, page 5, line 1-4, page 7, line 1-2, page 9, line11-15).

  1. What could be the reason for DHE staining even in control sample?

Response 5: Thanks very much for your comments. The fluorescence intensity of DHE in the control group was weak, which may be due to nonspecific staining. Moreover, there is stronger fluorescence intensity in the group of DM. And the results of additional experiments with DCFH staining and MitoSOX staining showed a relative weak fluorescence intensity in the control group.

  1. Authors need to perform more mitochondrial related parameters if they are focused on SIRT3.

Response 6: Thanks very much for your comments. According to your great suggestion, mtDNA copy number has been performed. The related methods, results and discussion have been revised (page 7, line 5-11, page 8, line 1-6, page 9, line 12-15).

  1. Did the authors evaluate changes in nitric oxide expression? and particularly, nitrative stress markers? 

Response 7: Thanks very much for your comments. The improvement of endothelial function suggested the (nitric oxide) NO production was increased. However, changes in NO expression and nitrative stress had not been measured in the present study. The related information has been added into discussion in the revised MS (page 8, line 26-28).

Reviewer 2 Report

Aim of the study is to investigate the role of dihydromyricetin (DHY) on vascular endothelial dysfunction in diabetic mice. The authors showed that DHY improved endothelium-dependent relaxation of thoracic aorta, inhibited oxidative stress and enhanced SIRT3 and SOD2 protein expression in thoracic aorta of diabetic mice. Moreover, they demonstrated that DHY improved endothelial dysfunction in a SIRT3-dependent manner.

The study is of interest, clear and well written. However, I have some comments.

In particular:

- Pag 3, line 4: the sentence “during the whole experiments” is repeated. Please delete it. - Pag 6, line 18: the abbreviation “WT” is not defined. Please check all the abbreviations in the text. - Pag 9, lines 1-2: the sentence “The diabetes morbidity is increasing with years, which has become a worldwide public health problem threatening to human health” is not well written, please rewrite it. - Pag 9: the discussion about the role of DHY and atherosclerosis could be amplified (Yang D, Yang Z, Chen L, et al. Dihydromyricetin increases endothelial nitric oxide production and inhibits atherosclerosis through microRNA-21 in apolipoprotein E-deficient mice. J Cell Mol Med. 2020;24(10):5911-5925. doi:10.1111/jcmm.15278; Huang B, Li Y, Yao Y, Shu W, Chen M. Dihydromyricetin from ampelopsis grossedentata protects against vascular neointimal formation via induction of TR3. Eur J Pharmacol. 2018;838:23-31. doi:10.1016/j.ejphar.2018.09.002). - Pag 9, line 33: finally, it would be interesting to amplify the discussion about the role of DHY and ischemia/reperfusion injury (Liu S, Ai Q, Feng K, Li Y, Liu X. The cardioprotective effect of dihydromyricetin prevents ischemia-reperfusion-induced apoptosis in vivo and in vitro via the PI3K/Akt and HIF-1α signaling pathways. Apoptosis. 2016;21(12):1366-1385. doi:10.1007/s10495-016-1306-6).

Author Response

Response to Reviewer 2 Comments

English language and style

( ) Extensive editing of English language and style required  
( ) Moderate English changes required  
(x) English language and style are fine/minor spell check required  
( ) I don't feel qualified to judge about the English language and style 

Thanks very much for your comments. We have double checked the MS to avoid any typographical and grammar mistakes. The MS has also been read and altered by a native English speaker and we hope that it is improved. All the corrections have been highlighted in red in the revised MS.

Yes

Can be improved

Must be improved

Not applicable

Does the introduction provide sufficient background and include all relevant references?

(x)

( )

()

( )

Is the research design appropriate?

(x )

()

()

( )

Are the methods adequately described?

(x )

()

( )

( )

Are the results clearly presented?

(x)

( )

()

( )

Are the conclusions supported by the results?

(x)

()

( )

( )

Thanks very much for your comments.

Comments and Suggestions for Authors

Aim of the study is to investigate the role of dihydromyricetin (DHY) on vascular endothelial dysfunction in diabetic mice. The authors showed that DHY improved endothelium-dependent relaxation of thoracic aorta, inhibited oxidative stress and enhanced SIRT3 and SOD2 protein expression in thoracic aorta of diabetic mice. Moreover, they demonstrated that DHY improved endothelial dysfunction in a SIRT3-dependent manner.

The study is of interest, clear and well written. However, I have some comments.

In particular:

- Pag 3, line 4: the sentence “during the whole experiments” is repeated. Please delete it.

Response: Thanks very much for your comments. We are sorry for the mistake and have deleted it.

- Pag 6, line 18: the abbreviation “WT” is not defined. Please check all the abbreviations in the text.

Response: Thanks very much for your comments. We have defined the abbreviation of WT in the revised MS and we have double checked all the abbreviations in the text to avoid any loss of definition.

- Pag 9, lines 1-2: the sentence “The diabetes morbidity is increasing with years, which has become a worldwide public health problem threatening to human health” is not well written, please rewrite it.

Response: Thanks very much for your comments. The sentence has been changed as “The diabetes with an increasing morbidity has become a worldwide public problem threatening to human health”. (page 8, line 8-9)

- Pag 9: the discussion about the role of DHY and atherosclerosis could be amplified (Yang D, Yang Z, Chen L, et al. Dihydromyricetin increases endothelial nitric oxide production and inhibits atherosclerosis through microRNA-21 in apolipoprotein E-deficient mice. J Cell Mol Med. 2020;24(10):5911-5925. doi:10.1111/jcmm.15278; Huang B, Li Y, Yao Y, Shu W, Chen M. Dihydromyricetin from ampelopsis grossedentata protects against vascular neointimal formation via induction of TR3. Eur J Pharmacol. 2018;838:23-31. doi:10.1016/j.ejphar.2018.09.002).

Response: Thanks very much for your comments. The discussions about the role of DHY and atherosclerosis as well as ligation-induced carotid artery neointimal formation have been added into the revised MS (page 8, line 20-21).

- Pag 9, line 33: finally, it would be interesting to amplify the discussion about the role of DHY and ischemia/reperfusion injury (Liu S, Ai Q, Feng K, Li Y, Liu X. The cardioprotective effect of dihydromyricetin prevents ischemia-reperfusion-induced apoptosis in vivo and in vitro via the PI3K/Akt and HIF-1α signaling pathways. Apoptosis. 2016;21(12):1366-1385. doi:10.1007/s10495-016-1306-6).

Response:Thanks very much for your comments. The role of DHY and ischemia/reperfusion injury has been added into the discussion in the revised MS (page 9, line 19-21).

Round 2

Reviewer 1 Report

Accept in present form